# A Numerical Case Study of Particle Flow and Solar Radiation Transfer in a Compound Parabolic Concentrator (CPC) Photocatalytic Reactor for Hydrogen Production

Jiafeng Geng [1,2], Qingyu Wei [3], Bing Luo [4], Shichao Zong [1,2], Lijing Ma [4], Yu Luo [1,2,*], Chunyu Zhou [1,2,*] and Tongkun Deng [2]

[1] Key Laboratory of Subsurface Hydrology and Ecological Effects in Arid Region of the Ministry of Education, Chang'an University, Xi'an 710054, China; gengjf@chd.edu.cn (J.G.); shichaozong@chd.edu.cn (S.Z.)
[2] School of Water and Environment, Chang'an University, Xi'an 710054, China; deng17870061003@outlook.com
[3] Beijing Aerospace Propulsion Institute, Beijing 100191, China; wei.1203@stu.xjtu.edu.cn
[4] International Research Center for Renewable Energy, State Key Laboratory of Multiphase Flow in Power Engineering, Xi'an Jiaotong University, Xi'an 710049, China; luobing@xjtu.edu.cn (B.L.); ljma@mail.xjtu.edu.cn (L.M.)
* Correspondence: 13289367808@163.com (Y.L.); zhouchy@chd.edu.cn (C.Z.); Tel.: +86-02982339965 (Y.L. & C.Z.)

**Highlights:**

**What are the main findings?**

- A comprehensive simulation model including particle flow and radiation transfer was developed for a CPC photocatalytic reactor.
- The ray tracing method was utilized to determine the radiation reaching the surface of the receiving tube, while the discrete ordinates method (DOM) was also employed to solve the radiative transfer equation (RTE), which shows the complete process of solar energy transfer.

**What is the implication of the main finding?**

- Local volume radiative power absorption (LVRPA) and total radiative power absorption (TRPA) inside the receiving tube was obtained by this study, which is critical data for the photocatalytic reactor.
- Natural convection with intermittent disturbances is demonstrated to be effective operating mode for the CPC photocatalytic reactor.

**Abstract:** Compound parabolic concentrator (CPC) photocatalytic reactors are commonly used for photocatalytic water splitting in hydrogen production. This study aimed to gain a better understanding of the physical processes in CPC photocatalytic reactors and provide theoretical support for their design, optimization, and operation. The analysis involved the ray tracing approach, Euler–Euler two-fluid model, and discrete ordinates method (DOM) to study solar radiation transfer and particle flow in the reactor. The distribution of solar radiation on the receiving tube's surface after CPC concentration was obtained by conducting the ray tracing approach. This solar radiation distribution was then coupled into the Euler–Euler two-fluid model to solve for the natural convection flow field, the temperature field, and particle phase volume fraction distribution inside the receiving tube over a period of 120 s. Lastly, the discrete ordinates method (DOM) was used to analyze the transfer of radiation inside the receiving tube at different times, obtaining the distribution of local volume radiative power absorption (LVRPA) and the total radiative power absorption (TRPA) inside the tube. The results showed that the TRPA reached its maximum at 120 s, accounting for 66.61% of the incident solar UV radiation. According to the above results, it could be suggested that adopting an intermittent operation mode in CPC photocatalytic reactors is reasonable and efficient.

**Keywords:** compound parabolic concentrator (CPC) photocatalytic reactor; natural convection; ray tracing method; Euler–Euler two-fluid flow model; radiative transfer equation

## 1. Introduction

Solar-driven photocatalysis is a promising route to produce green hydrogen [1]. However, its relatively low solar-to-hydrogen conversion efficiency severely restricts its industrial application [2]. The development of photocatalytic technology mainly includes two aspects. On the one hand, it is the development of efficient visible-light-driven photocatalyst, and, on the other hand, it is the development of high-performance photocatalytic reactors [3]. While the former has achieved significant breakthroughs over decades of development [4–7], the latter one is relatively insufficient [8,9], which may be one of the main reasons limiting the industrial application of this technology [10].

Generally speaking, in photocatalytic reactors that use direct sunlight as the light source, concentrators are commonly employed [11]. This is not only because it is necessary to increase light intensity to enhance reaction rates but also because the thermal effects after concentration can facilitate the progress of the reactions [12]. In photocatalysis technology, there are two commonly used concentrators: one is the parabolic trough collector (PTC) [13], and the other is the compound parabolic concentrator (CPC) [14]. Despite the higher concentration ratio of the PTC concentrator, it can only utilize direct solar radiation and cannot collect scattered solar radiation. Moreover, it requires a high-precision tracking system which is complex and costly, significantly limiting its scaling up [15], whereas the CPC concentrator can utilize both direct and scattered radiation and without needing a tracking system or requiring only a low-precision one. Therefore, CPC has greater potential for solar utilization and conversion in industrial applications [16]. Based on the characteristics mentioned above, CPC photocatalytic reactors have been well applied in photocatalytic water treatment and purification [17–20]. However, the application of CPC photocatalytic reactors in the field of photocatalytic hydrogen production from water decomposition was first reported by Jing et al. from the State Key Laboratory of Multiphase Flow in Power Engineering (SKMFPE) [21]. Based on this research, our group has further investigated the scaling up of CPC photocatalytic reactors, of which a landmark achievement is the construction of a pilot plant near Xi'an with a light area of 103.7 m$^2$ [22].

As the research further develops, we gradually realize that the issues of particle flow and radiation transfer in CPC photocatalytic reactors are two crucially important theoretical issues. Understanding these issues can guide the design and operation of CPC photocatalytic reactors. Therefore, we have conducted a large number of experiments and simulations focusing on these two issues. Regarding the particle flow, Hu et al. first studied the solid–liquid flow characteristics in CPC photocatalytic reactors by introducing an algebraic slip mixture (ASM) model [23]. Subsequently, Geng et al. conducted experimental research on the distribution and transfer behavior of particles inside a photocatalytic reactor, revealing the particle velocity distribution and particle size distribution of the photocatalyst suspension system [24,25]. Additionally, on account of the second issue, namely the radiation transfer problem within the reactor, we also conducted in-depth research. Yang et al. developed a surface uniform concentrator (SUC) through ray tracing to solve the non-uniform light radiation on the surface of receiving tube caused by CPC concentrators [26]. Subsequently, Wei et al. conducted experimental studies based on SUC [27]. Yang et al. further utilized particle concentration distribution calculated by computational fluid dynamics (CFD) to modify the Six Flux Model (SFM), which was used to solve the radiation distribution within the receiving tube [28]. This was a significant attempt that considered the influence of particle flow on radiation transfer. Furthermore, Hou et al. coupled the particle concentration distribution calculated with CFD with Monte Carlo methods to further optimize the calculation of the radiation field in the photocatalytic reactor [29]. From the above research, it is evident that as the research progresses, we gradually realized the close interactions between particle flow and radiation transfer, necessitating coupled analysis and the establishment of comprehensive simulation models for research and analysis. Ren et al. were the first to initiate research in this direction. Based on previous studies, they used P1 approximation to solve the radiative transfer equation (RTE), thus developing a comprehensive simulation model for CPC photocatalytic

reactors [30]. Wei further refined this model by using the SFM to solve the radiation distribution more accurately inside the receiving tube [31].

Based on the relevant research on CPC photocatalytic reactors conducted by the group from SKMFPE, in this study, a comprehensive simulation model that includes particle flow and radiation simulation was proposed, also followed by a case study. With the ray tracing method, the distribution of solar radiation on the surface of the receiving tube was calculated using ray tracing, with specific parameters set for Xi'an at 12:00 p.m. on 1 July in the summer. Furthermore, this study utilized the Euler–Euler two-fluid model to solve the natural convection flow field and temperature field inside the receiving tube, which also coupled the radiation distribution on the surface of the receiving tube as the boundary heat source. Finally, the discrete ordinates method (DOM) was employed to solve the RTE and obtain the local volume radiative power absorption (LVRPA) and total radiative power absorption (TRPA) inside the receiving tube. In fact, the DOM method offers higher precision in solving the radiative transfer equation (RTE) than the SFM, but it also consumes more computational resources [32]. In this study, to address this issue, a time decoupling approach was applied to the comprehensive model. Specifically, the data from the simulation results of particle flow at specific time points were extracted and then coupled into the DOM for solving, rather than conducting simultaneous transient calculations. The results obtained from this case study led to the conclusion that adopting a mode of operation combining natural convection with intermittent disturbances is reasonable for CPC photocatalytic reactors. This conclusion provides valuable insights for the operation, simulation, and optimization of CPC photocatalytic reactors.

## 2. Results and Discussion

### 2.1. The Distribution of Radiation on the Surface of Receiving Tube

Figure 1 presents the simulation results of solar radiation energy distribution at 12:00 p.m. on 1 July in Xi'an. Figure 1a illustrates the direction of direct solar radiation at that time via a thick red arrow. It can be observed from the figure that the sun's position at this moment is slightly east of due south, indicating an azimuth angle of less than 180°. Consequently, the oblique incidence state relative to the CPC concentrator leads to an uneven distribution of surface light radiation energy received by the reaction tube. As depicted in Figure 1b,c, a region of relatively strong radiation is noticeable on the side of the incident light, particularly the lower part of the tube on the east side. Within this area, the radiation intensity on the entire eastern half of the pipe surface is relatively high, while the optical radiation intensity on the west side of the pipe wall surface is low. Further analysis of the surface radiation reveals that the maximum irradiation power on the tube wall reaches 7381.0 $W \cdot m^{-2}$, which is approximately 7.6 times the incident solar radiation intensity (965.5 $W \cdot m^{-2}$). The average irradiation intensity on the tube wall is 750.07 $W \cdot m^{-2}$, which is 0.78 times the incident solar radiation intensity. This discrepancy arises due to the non-uniform solar radiation distribution on the surface of the tube wall, with certain areas such as the bottom of the receiving tube not receiving effective solar radiation. The total amount of irradiation received on the surface of the receiving tube is 188.51 W, while the total amount of solar radiation incident from the opening surface of the CPC concentrator is 596.46 W. Consequently, the light collection efficiency of the CPC concentrator is calculated to be 31.6%. The lower concentration efficiency in the output is primarily attributed to the use of a low-precision tracking system in the CPC system. In the experimental setup of this study, the adjustment rate is 5° per hour [22]. For more detailed information on this issue, please see the Supplementary Materials.

### 2.2. Natural Circulation Flow in the Receiving Tube

From Section 2.1, it could be observed that due to the uneven distribution of solar radiation energy received on the surface of the receiving tube, temperature, and density of the fluid in the receiving tube are different, which cause natural convective flow. This flow can disturb the photocatalyst particles, aiding in maintaining their suspended state.

Figure 2 illustrates the temperature distribution inside the receiving tube. In this study, the temperature variation and flow field changes inside the receiver tube were computed from the initial state with heating for 120 s, assuming that at the initial state, the fluid temperature inside the receiving tube is 303 K, which is equal to the ambient temperature. It can be observed from Figure 2 that, as the fluid inside the tube heats up from initial 303 K to above 308 K in 120 s, the temperature distribution in the receiving tube exhibits non-uniform distribution due to the non-uniform solar radiation received on the surface of the receiving tube. Therefore, on the side of the receiving tube where the solar radiation intensity is stronger, i.e., the eastern side of the tube, the fluid temperature is significantly higher than on the other side. Additionally, at the cross-section of the tube at L = 0.2 m, it is noticeable that the lower-right part of the tube shows higher fluid temperatures at all the times, which is consistent with the distribution of solar radiation intensity. Furthermore, the heated region gradually expands from the boundary towards the center of the tube over time. At the cross-sections of the pipeline at L = 0.6 m, 1.0 m, and 1.4 m, it can be observed that the central-right part of the tube shows higher fluid temperatures at different times. Comparing with the situation at L = 0.2 m, it can be seen that the temperature distribution at L = 0.6 m, 1.0 m, and 1.4 m does not completely align with the distribution of solar radiation intensity, indicating potential influence from the natural convection flow field on the temperature distribution. At the cross-section of L = 1.8 m, the fluid temperature gradually increases over time, but the temperature distribution tends to be more uniform, and the overall temperature is lower, ranging between 304 and 305 K.

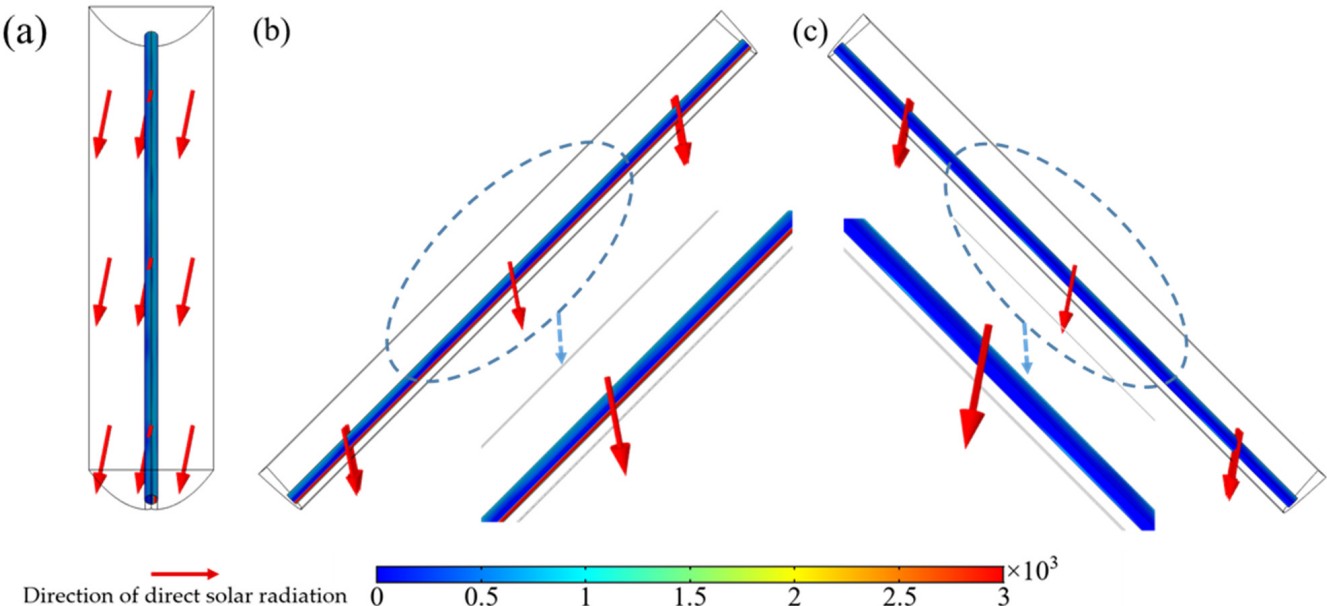

**Figure 1.** (**a**) Top view, (**b**) right view, and (**c**) left view of the contour of the radiation distribution on the surface of the receiving tube, respectively.

Figure 3 illustrates the natural circulation flow field in the receiving tube. It can be observed from Figure 3 that magnitude of the velocity of natural circulation flow inside the tube gradually increases over time. At t = 120 s, the maximum magnitude of the velocity inside the tube can reach 0.43 m/s. Regarding the direction of the velocity vectors, at different times, the directions of fluid flow at L = 0.2 m, 0.6 m, 1.0 m, and 1.4 m cross-sections are similar, with the fluid in the top of the tube flows along the axis ranging from L = 0.2 m to L = 1.4 m, while the fluid in the bottom of the tube flows along the axis ranged from L = 1.4 m to L = 0.2 m. This is mainly because the lower-right region of the tube possesses higher fluid temperatures, resulting in the heated fluid to first flow upward to the top on the cross-section, and then flow upward along the axis of the tube. In contrast, the fluid in the top of the tube has lower temperatures due to lower solar radiation intensity.

Therefore, the fluid initially flow from the top to the bottom on the cross-section, and then flows downward along the axis of the tube, which forms an overall circulation flow inside the receiving tube. Additionally, two symmetrical vortices appear on the cross-section of the tube, indicating that the flow on the cross-section is also a circulating flow. The fluid flows from the bottom in both sides of the tube to the top in central of the tube, and then from the top in central of the tube to the bottom. Furthermore, the circulation flow velocity of the fluid reaches its maximum value at L = 1.8 m cross-section. Regarding the direction of the velocity vectors, the overall direction of the flow is consistent with other cross-sections, but the magnitude of velocity is larger, and the direction of the velocity vectors appears more chaotic. This indicates that the fluid at the L = 1.8 m cross-section is well mixed, which also explains the phenomenon of a relatively uniform temperature distribution at the top in the previous paragraph.

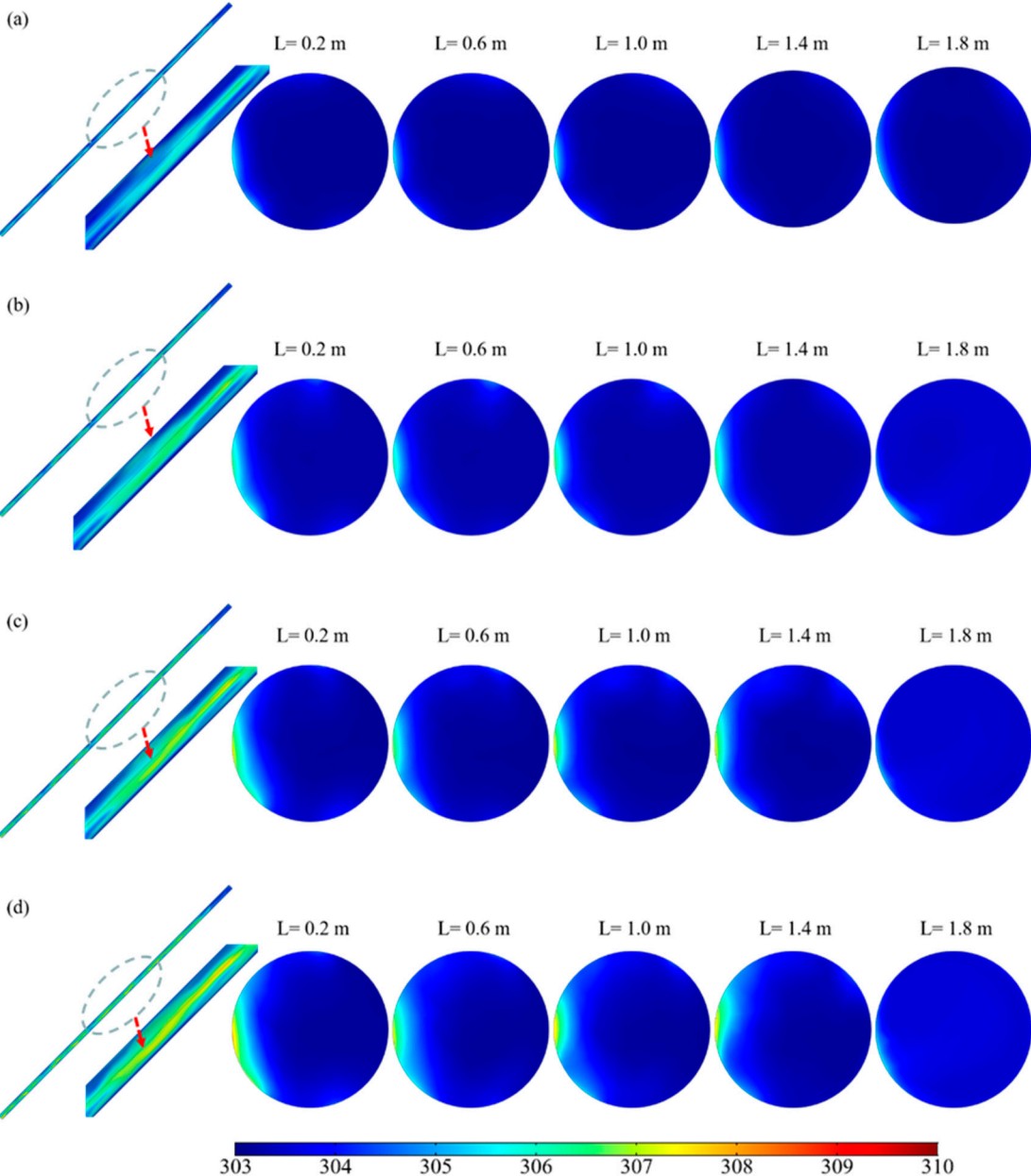

**Figure 2.** Contour of temperature of L = 0.2 m, 0.6 m, 1.0 m, 1.4 m, and 1.8 m cross-sections in receiving tube at (**a**) t = 30 s, (**b**) t = 60 s, (**c**) t = 90 s, and (**d**) t = 120 s.

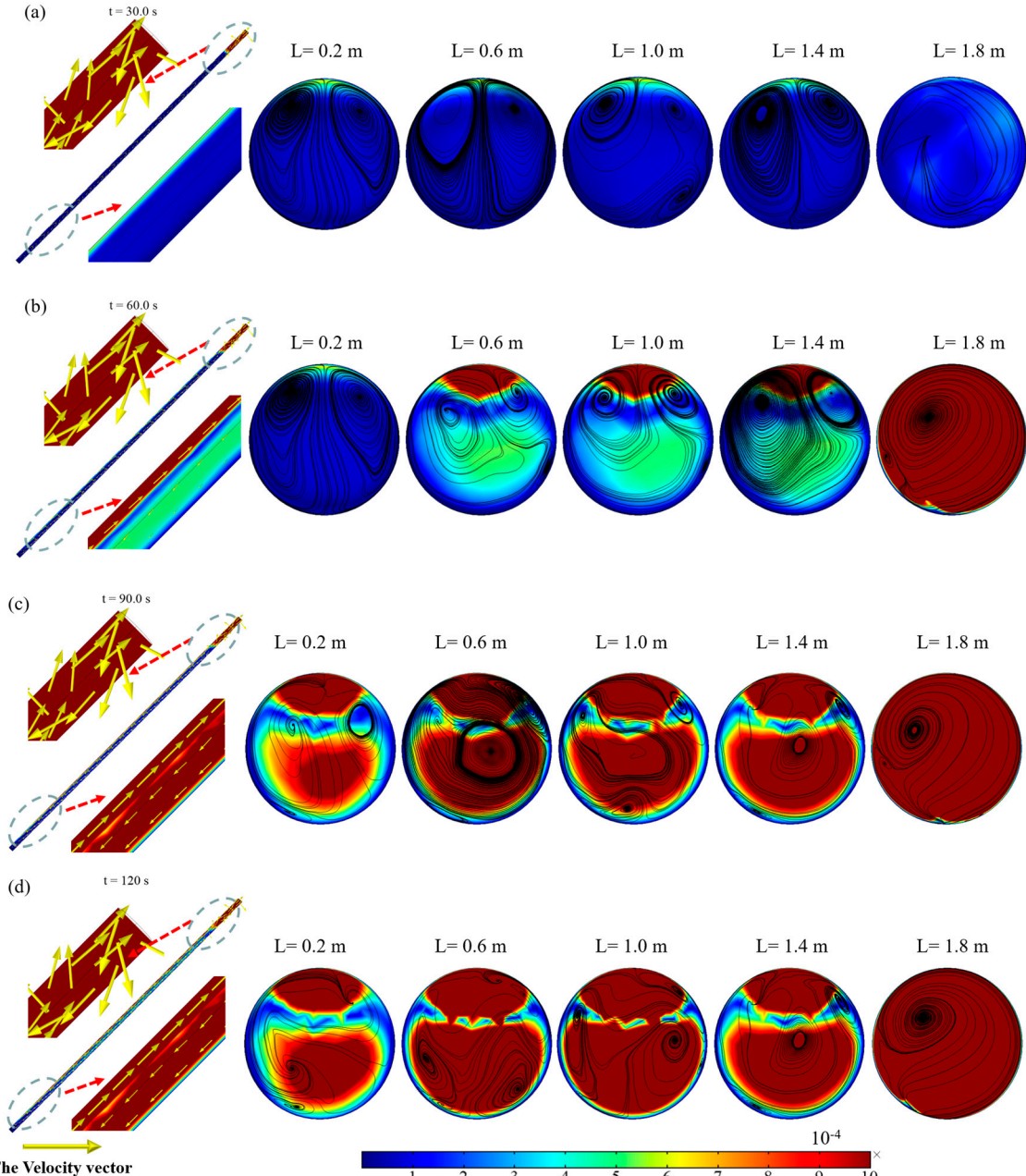

**Figure 3.** Contours, streamlines, and vectors of the natural convective flow of L = 0.2 m, 0.6 m, 1.0 m, 1.4 m, and 1.8 m cross-sections in receiving tube at (**a**) t = 30 s, (**b**) t = 60 s, (**c**) t = 90 s, and (**d**) t = 120 s.

### 2.3. The Phase Volume Fraction of the Photocatalysts in the Receiving Tube

As shown in Figure 4, the distribution of the photocatalyst particles phase fraction inside the receiving tube varies over time, assuming that at the initial condition, the liquid inside the receiving tube is a homogeneous suspension with a volume fraction of $2.5 \times 10^{-4}$, which is based on the actual load of photocatalyst [33]. The figure illustrates that over time, the distribution of the particles phase fraction inside the tube gradually changes from a uniform state to a segregation state. On the one hand, the variation in the particle phase fraction at cross-sections L = 0.2 m, 0.6 m, 1.0 m, and 1.4 m of the receiving tube appears to be quite similar over time. It is noticeable that the particle phase fractions at the top of the cross-sections tend to approach zero, while particle phase fractions at the bottom of the cross-sections increase gradually. This phenomenon suggests that the natural circulation flow inside the receiving tube is insufficient to ensure that the particles remain

suspended. On the other hand, at the L = 1.8 m cross-section, after a period of time, the particle phase fraction notably decreases with time, almost reaching zero at t = 120 s. As the photocatalyst particles settle inside the tube, the changes in the distribution of the particle phase volume fraction greatly affect the absorption of light radiation via the suspended particle system inside the tube. The following section will focus on the changes in the radiation transfer characteristics within the receiving tube.

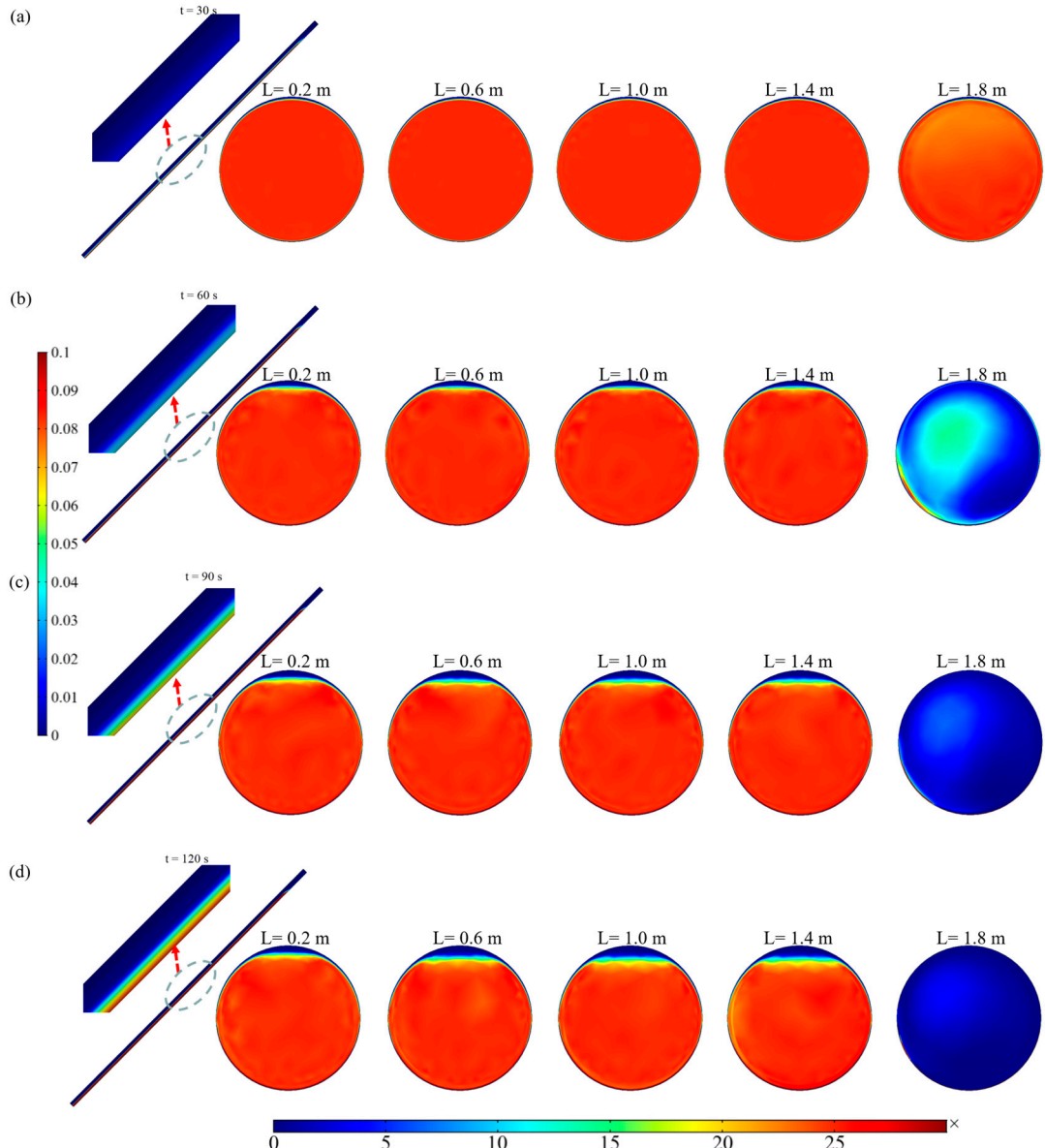

**Figure 4.** Contours of the phase volume fraction of photocatalyst at L = 0.2 m, 0.6 m, 1.0 m, 1.4 m, and 1.8 m cross-sections in receiving tube at (**a**) t = 30 s, (**b**) t = 60 s, (**c**) t = 90 s, and (**d**) t = 120 s.

### 2.4. Radiation Transfer Characteristics in the Receiving Tube

According to the analysis in Section 2.3, the particle phase volume fraction distribution is relatively similar in the different cross-sections within the receiving tube, except for the L = 1.8 m cross-section. Therefore, in this section, the cross-section at L = 1.0 m is selected to discuss the influence of particle distribution on light radiation transmission. Firstly, the absorption coefficient is related to the particles phase volume fraction, as illustrated in Figure 5a,b; the absorption coefficient in the bottom of the cross-section of the receiving tube is larger than in the top at all the times except for t = 0 s, due to the

larger particle phase volume fraction in the bottom of the cross-section caused by particle segregation phenomenon. Furthermore, this characteristic deeply affects the absorption of solar radiation via photocatalyst particles inside the receiving tube. To assess the impact of increased particle phase volume fraction on solar radiation absorption, the local volume radiative power absorption (LVRPA) was employed. As shown in Equation (1), LVRPA for each wavelength is determined by integrating the radiation intensity over the entire $4\pi$ space and multiplying by the absorption coefficient.

$$\text{LVRPA}_\lambda = \kappa_\lambda^* c_{cat} \int_0^{4\pi} I_\lambda d\Omega \tag{1}$$

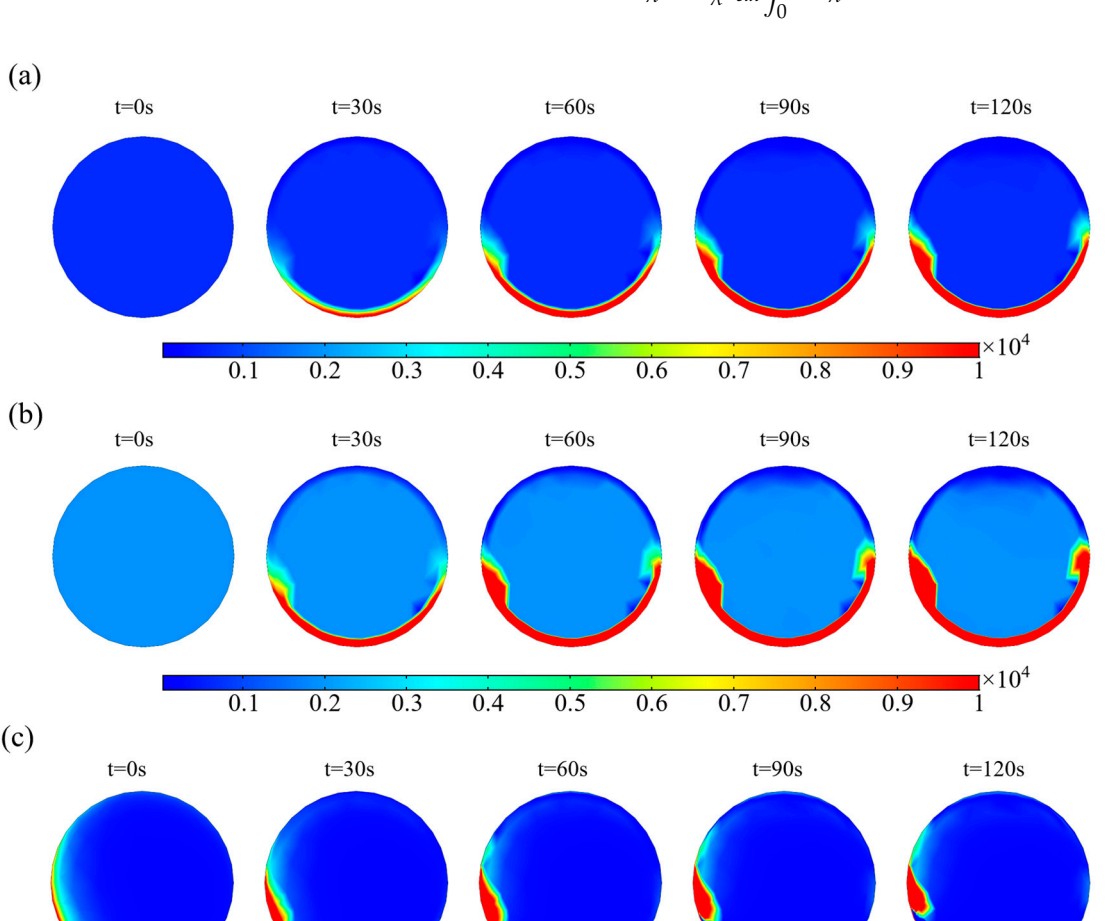

**Figure 5.** Contours of (**a**) absorption coefficient of the UVA spectrum and (**b**) absorption coefficient of the UVB spectrum. (**c**) LVRPA in the receiving tube of L = 1.0 m cross-section in receiving tube at t = 30 s, t = 60 s, t = 90 s, and t = 120 s, respectively.

Figure 5c illustrates the absorption coefficient and local volume radiative power absorption (LVRPA) of the photocatalyst suspension in the receiving tube. Figure 5c depicts the distribution of LVRPA for the photocatalyst suspension, which describes the local absorption radiation via the photocatalyst in the receiving tube. Therefore, the LVRPA is not only related to the photocatalyst phase volume fraction distribution but also to the radiation distribution. It was discussed in Section 2.1 that the distribution of solar radiation received by the receiving tube is non-uniform on the surface. Thus, it could be seen that the LVRPA exhibits a non-uniform distribution at t = 0, even if the particle phase volume fraction at this time is uniform. Specifically, the LVRPA on the lower-right part of the receiving tube exhibits significantly larger than other regions. As time goes on, although

the position of this region displays no obvious change, the area and shape of it change, which gradually reduces it into a smaller area and becomes deeper in radial direction. Furthermore, the magnitude of the LVRPA increases over time, which is assumed to be related to the segregation process. This phenomenon indicates that the higher particle phase volume fraction caused by the segregation process at the bottom of the receiving tube may facilitate the absorption of solar radiation.

The total radiative power absorption (TRPA) corresponds to the power absorbed in the receiving tube, and it was calculated by integrating the LVRPA over the volume:

$$\text{TRPA} = \int_V \text{LVRPA}_\lambda \, dV \qquad (2)$$

The blue diagonal bar chart in Figure 6 represents the TRPA inside the receiving tube. According to the calculations in Section 3.1, the total radiation received on the surface of the receiving tube is 188.51 W. However, the photocatalyst used in this study was assumed to be P25 TiO$_2$, which is a typical photocatalyst, and its major absorption spectrum are UVA radiation (about 6.3% of the solar spectrum energy) and UVB radiation (about 1.5% of the solar spectrum energy). Therefore, the actual radiation absorbed by the photocatalyst amounts to 12.25 W. Based on this value, the ratio of the TRPA to the total received UV spectrum of the solar radiation could be derived, as depicted by the red diagonal bars in Figure 6 It can be observed that the TRPA value gradually increases over time and reaches its maximum value at t = 120 s within the simulated period. Combining the particle phase volume fraction distribution obtained in Section 2.3 and the discussion in the previous paragraph, we hypothesized that this result is due to the partial settling of particles at 120 s, resulting in a larger particle phase volume fraction at the bottom of the receiving tube. Consequently, the absorption coefficient in this area significantly increases, leading to an increase in the absorption of the solar radiation. This result also indicates that a certain degree of photocatalyst segregation process contributes to the enhancement of solar radiation absorption in the CPC tubular photocatalytic reactor. Therefore, in the application of such photocatalytic reactors, employing an operating mode of natural circulation supplemented with periodic disturbances, such as forced circulation, is rational and efficient.

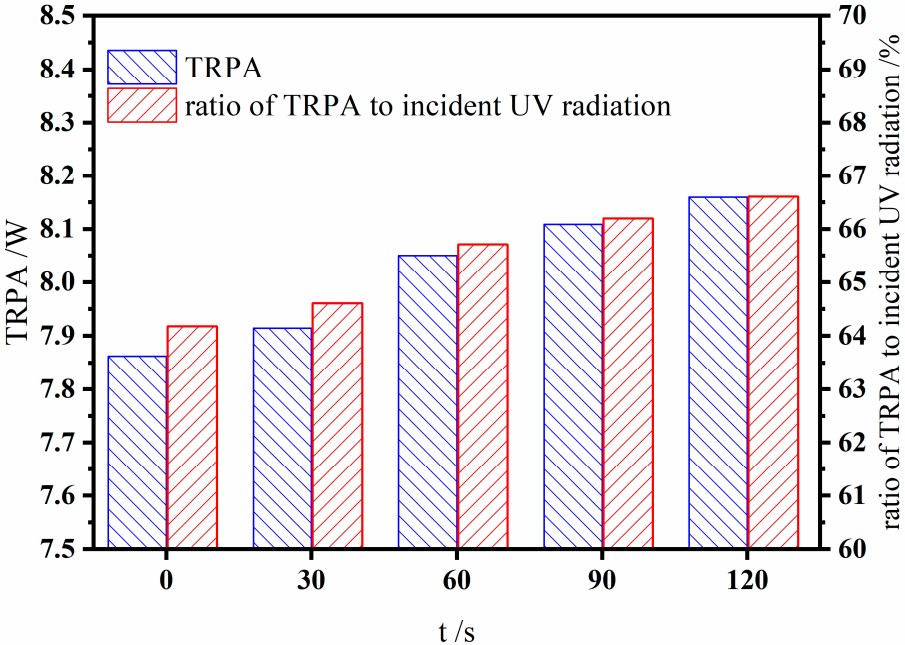

**Figure 6.** Bar chart of TRPA and its ration to incident UV radiation.

### 3. Mathematical Model

The basic parameters of the CPC tubular photocatalytic reactor are shown in Table 1, where the geometric parameters of the CPC concentrator and receiver tube are derived from a previously optimized design of a large-scale solar photocatalytic hydrogen production system, as detailed by Wei et al. [22], and the geometry details of the CPC were elaborated in the supporting information. The three-dimensional model established in this study is shown in Figure 7. The direction of solar radiation in the figure is determined based on the actual position of the sun. After being reflected by the CPC concentrator, solar radiation converges onto the surface of the receiver tube. Among these radiations, a part of ultraviolet and visible light energy is absorbed by the semiconductor photocatalyst, thereby exciting the semiconductor material to generate electron–hole pairs. Another part of the longer visible wavelength and infrared light energy is absorbed by the fluid and converted into thermal energy, thereby raising the temperature of the fluid inside the reactor. In this process, different regions of the fluid receive different amounts of heat, resulting in a non-uniform temperature distribution and density difference, thus forming natural convection flow inside the tube. Before performing calculations, it is necessary to correspond the Cartesian coordinate system used in the simulation with the actual directions. In this study, the positive direction of the $x$-axis corresponds to the due north direction, and the positive direction of the $y$-axis corresponds due west. The overall layout of the photocatalytic reactor discussed in this study is oriented in the north–south direction, with the inclined direction from south to north, and the northern end is higher. The inset in Figure 1 shows a cross-sectional schematic of the CPC concentrator and the receiving tube. The compound parabolic surface can concentrate the incident solar radiation onto the reactor tube wall.

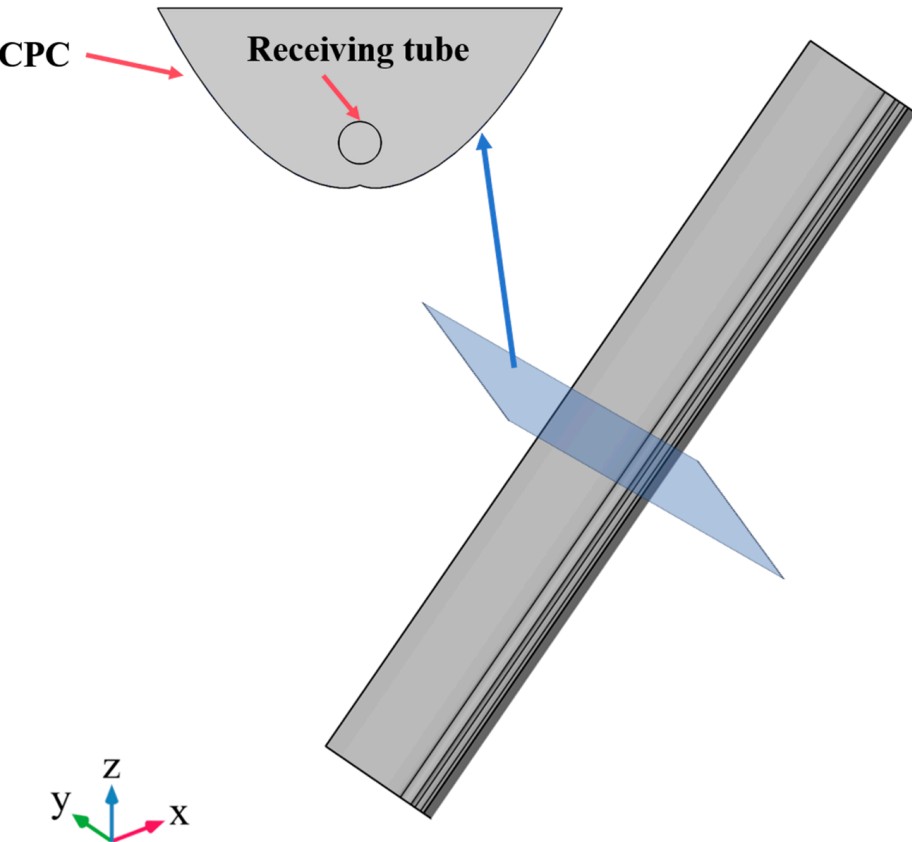

**Figure 7.** The geometric model of CPC tubular photocatalytic reactor.

**Table 1.** List of the parameters of CPC tubular photocatalytic reactor.

| Parameters | Numerical Value |
| --- | --- |
| $d_R$/mm | 30 |
| $l_R$/m | 2 |
| $r_B$/mm | 15 |
| Gap/mm | 25 |
| $\theta a$/° | 5 |
| $\theta t$/° | 60 |

The computational grid used in this study was generated by sweeping. Therefore, suitable plane grids are first constructed in the section and are then stretched along the axis, as shown in Figure 8. Since the simulation involves two fundamental processes—the transmission of light rays between the receiving tube and the CPC concentrator, and the flow and heat transfer inside the reaction tube—different methods are adopted for grid partitioning in the two regions. For the space between the reaction tube and the concentrator surface, the main task is to solve the ray tracing equation, so a relatively sparse grid is partitioned. The plane grid adopts quadrilateral meshes, and the swept grid becomes hexahedral meshes. This grid partitioning strategy can significantly reduce the number of grids without affecting accuracy. Regarding the internal region of the reaction tube, due to the involvement of complex flow, heat transfer, and radiation transfer issues, an unstructured triangular mesh partitioning strategy is adopted on the section. Moreover, boundary layer refinement is performed near the pipe wall, with three layers of boundary layer grid sets. The final grid partitioning results in a total of 65,050 grids.

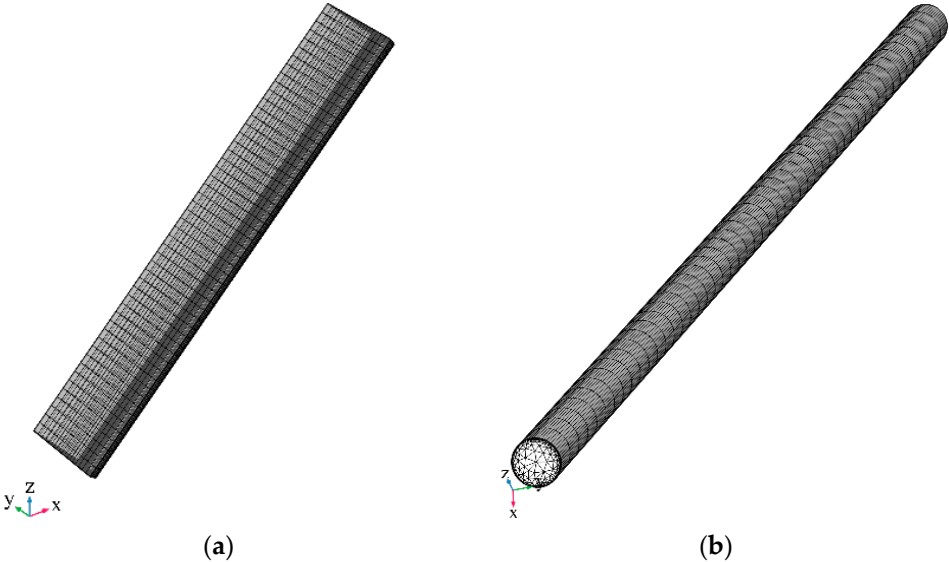

        (**a**)                                    (**b**)

**Figure 8.** Schematic of the grids of the (**a**) complete region and (**b**) receiving tube region.

### 3.1. Physical Model

The physical model established in this study is constructed based on the operation of the reactor under direct solar radiation, including the processes of solar radiation transmission to the surface of the CPC concentrator and the receiving tube, two-phase flow and heat transfer of the photocatalyst suspension in the receiving tube, and the transmission of radiant energy in the medium inside the receiving tube. Therefore, the ray tracing approach, Euler–Euler two-fluid flow model, and discrete ordinates method (DOM) were applied to solve the issue.

### 3.1.1. Ray Tracing Approach

In this study, the ray tracing model was applied to model the radiation transfer process from the sunlight to the surface of the CPC concentrator and the receiving tube. The ray tracing model assumes that the geometric dimensions of the model are sufficiently large, such that even the smallest details in the geometry are much larger than the wavelength of light; thus, diffraction effects are not produced, and the wave nature of light is not considered. According to the classical field theory, the ray tracing equations for sunlight can be expressed as first-order differential equations with respect to the wave vector $\mathbf{k}$ and the position vector $\mathbf{q}$, as outlined in Equation (3):

$$\begin{cases} \frac{d\mathbf{k}}{dt} = 0 \\ \frac{d\mathbf{q}}{dt} = \frac{c\mathbf{k}}{n|\mathbf{k}|} \end{cases} \tag{3}$$

In this study, the light source is set according to real solar radiation data. The selected location is Xi'an City (latitude 34.27° N, longitude 108.90° E), with a date of 1 July. The incident solar radiation is set to 965.5 W·m$^{-2}$ according to the experiments reported in the literature regarding the average summer solar radiation in Xi'an City [22]. It should be noted that solar radiation intensity is significantly influenced by weather conditions, and, in this case, the analysis was conducted under relatively high solar radiation intensity in the summer, which can fully reflect the hydrogen production performance of the system. In relation to the boundary conditions, 90% of the light is set as specular reflection on the CPC surface, while 10% of the radiant energy is absorbed. For the receiving tube, all light entering the tube wall was set to be absorbed.

### 3.1.2. Euler–Euler Two-Fluid Flow Model

As there are a certain amount of photocatalyst particles suspending inside the receiving tube, the flow inside the tube is liquid–solid two-phase flow. According to previous research, the flow of the photocatalyst suspension in the receiving tube is always natural convection flow that is caused by a temperature gradient in the receiving tube [22]. Considering the large size of the model established in this study, describing the particles separately would require a tremendous number of computational resources. Therefore, the Euler–Euler model was chosen for modeling of the liquid–solid flow in the receiving tube. In the Euler–Euler model, the particle phase is considered as another fluid phase, using the same governing equations as the fluid phase. This hypothetical fluid phase is also referred to as pseudo-fluid [34]. Based on the above assumptions, the continuity equation for the two phases is expressed as follows:

$$\frac{\partial}{\partial t}(\rho_c \phi_c) + \nabla \cdot (\rho_c \phi_c \mathbf{u_c}) = 0 \tag{4}$$

$$\frac{\partial}{\partial t}(\rho_d \phi_d) + \nabla \cdot (\rho_d \phi_d \mathbf{u_d}) = 0 \tag{5}$$

$$\phi_c = 1 - \phi_d \tag{6}$$

The momentum equation for the two phases can be expressed in the following form:

$$\rho_c \phi_c \left[ \frac{\partial}{\partial t}(\mathbf{u}_c) + \mathbf{u}_c \nabla \cdot (\mathbf{u}_c) \right] = \phi_c \nabla(-p\mathbf{I} + \mathbf{K_c}) + \phi_c \rho_c \mathbf{g} + \mathbf{F_{m,c}} + \phi_c \mathbf{F_c} \tag{7}$$

$$\rho_c \phi_c \left[ \frac{\partial}{\partial t}(\mathbf{u}_d) + \mathbf{u}_d \nabla \cdot (\mathbf{u}_d) \right] = \phi_d \nabla(-p\mathbf{I} + \mathbf{K}_d) + \phi_d \rho_d \mathbf{g} + \mathbf{F_{m,}}_d + \phi_d \mathbf{F}_d \tag{8}$$

where the viscous stress tensor **K** can be expressed in the following form based on the constitutive equation for Newtonian fluids:

$$\mathbf{K}_c = \mu_c^m \left( \nabla \mathbf{u_c} + (\nabla \mathbf{u_c})^T - \frac{2}{3} (\nabla \cdot \mathbf{u_c}) \mathbf{I} \right) \tag{9}$$

$$\mathbf{K}_d = \mu_d^m \left( \nabla \mathbf{u}_d + (\nabla \mathbf{u}_d)^T - \frac{2}{3} (\nabla \cdot \mathbf{u}_d) \mathbf{I} \right) \tag{10}$$

In this study, mixed viscosity assumption is employed, which means that the dynamic viscosity coefficients for the continuous and discrete phases are equal.

$$\mu_c^m = \mu_d^m = \mu_m \tag{11}$$

The mixed viscosity $\mu_m$ is typically obtained through empirical formulas or theoretical analysis models. In this study, the Krieger viscosity model [35] is used, as shown in Equation (12).

$$\mu_{mix} = \mu_c \left( 1 - \frac{\phi_d}{\phi_{d,\max}} \right)^{-2.5\phi_{d,\max}} \tag{12}$$

where $\varphi_{d,\max}$ represents the maximum solid phase volume fraction, which is commonly assumed to have a default value of 0.62.

Furthermore, in Equations (7) and (8), there is an interphase interaction force, $\mathbf{F_m}$. Although many forces contribute to it, in two-phase liquid–solid flow, the most important force among the interphase interaction forces is drag force. Therefore, this term is written as the drag force term, as follows:

$$\mathbf{F}_{drag,c} = -\mathbf{F}_{drag,d} = \beta \mathbf{u}_{slip} \tag{13}$$

where $\beta$ represents the drag coefficient; $\mathbf{u}_{slip}$ denotes the slip velocity, defined as $\mathbf{u}_{slip} = \mathbf{u}_c - \mathbf{u}_d$; and the drag coefficient $\beta$ can be expressed in the following equation:

$$\beta = \frac{3}{4d_\mathbf{p}} \phi_d C_D \rho_c \left| \mathbf{u}_{slip} \right| \tag{14}$$

where $d_\mathbf{p}$ represents the particle diameter; $C_D$ denotes the drag coefficient between fluid and particles. Regarding the particle diameter, as discussed in our previous research [24], photocatalyst particles tend to aggregate in the liquid, resulting in a particle size distribution. The average particle size D32 can be used as the hydraulic average particle size for calculation purposes, with a value of 3 μm. Since the volume fraction of the discrete phase in this study is relatively low, the Schiller–Naumann drag model was selected. This model is primarily derived from a rigid spherical single-particle derivation and is suitable for liquid–solid two-phase systems with low solid phase content.

$$C_D = \begin{cases} \frac{24}{\mathrm{Re}_p} \left( 1 + 0.15 \mathrm{Re}_p^{0.687} \right) & \mathrm{Re}_p < 1000 \\ 0.44 & \mathrm{Re}_p > 1000 \end{cases} \tag{15}$$

where $\mathrm{Re}_p$ is the Renolds number of photocatalyst particles, and the expression is

$$\mathrm{Re}_p = \frac{\rho_c \left| \mathbf{u}_{slip} \right| d_p}{\mu_c} \tag{16}$$

To establish the model of the natural circulation flow inside the receiving tube, the energy equation was introduced in the modeling process. Since the particle concentration

in the liquid–solid two-phase flow discussed in this study is relatively low, the following equation can be used:

$$\rho_c c_{p,c}\left(\frac{\partial T}{\partial t} + (\mathbf{u}_c \cdot \nabla)T\right) = -(\nabla \cdot \mathbf{q_h}) + \mathbf{K}_c : \mathbf{S}_c - \alpha_{p,c}\left(\frac{\partial p}{\partial t} + (\mathbf{u}_c \cdot \nabla)p\right) + Q \quad (17)$$

where the $\mathbf{q_h}$ represents the heat conduction term, for which the expression is $\mathbf{q} = -k\nabla T$.

With regard to the momentum and continuity equations, the boundary conditions are set as wall conditions for the bottom and the surroundings of the receiving tube. For only natural convection flow was considered, there is no inlet boundary in this study. The top of the receiving tube was set as a pressure outlet boundary and the pressure was set as the atmospheric pressure because, in the real system, the top of the receiving tube was opened for hydrogen discharge. Concerning the energy equation, the boundary conditions are set as the combination of solar radiation heating and convective heat dissipation. Among the two heat sources, convective heat could be expressed as Equation (18):

$$q_f = h(T_{air} - T) \quad (18)$$

where $T_{air}$ is the ambient temperature, with its value being 303 K in this study according to the average temperature in Xi'an in the summer.

Regarding the solar heat source, infrared light could be absorbed by the suspension in the receiving tube to generate heat, so the solar heat source can be described using Equation (19):

$$\frac{dq_{src}}{dt} = -\alpha_{ht}\sum_{j=1}^{N}\frac{\partial q_j}{\partial t}\delta(\mathbf{r} - \mathbf{q}_j) \quad (19)$$

where $q_{src}$ represents the heating source term in the fluid energy equation, $\alpha_{ht}$ denotes the medium's optical–thermal efficiency in the receiving tube. Considering that the infrared light in solar radiation is mainly absorbed by water, this coefficient is set to 0.173. $q_j$ represents the energy carried by the j-th ray.

It is worth noting that in this study, solar radiation is coupled as a boundary condition into the energy equation, allowing for the analysis of natural convection phenomena induced by solar radiation. This approach serves as a valuable complement to previous research efforts [28–31].

### 3.1.3. Discrete Ordinates Method

According to the theory of radiation transfer, incident light encountering particles may undergo absorption, transmission, and scattering. Based on the interaction between the medium and radiation, the radiation transfer equation (RTE) can be written as shown in Equation (20).

$$\begin{aligned} \mathbf{s} \cdot \nabla I(\mathbf{r}, \mathbf{s}) &= \varepsilon I_b(T) - (\kappa + \sigma)I(\mathbf{r}, \mathbf{s}) + S(\mathbf{r}, \mathbf{s}) \\ S(\mathbf{r}, \mathbf{s}) &= \frac{\sigma}{4\pi}\int_0^{4\pi} \phi_S(s, s')I(r, s')d\Omega' \end{aligned} \quad (20)$$

where S($\mathbf{r}$,$\mathbf{s}$) is the total energy scattered from any direction to the s-direction.

In this study, the discrete ordinates method (DOM) is employed to solve the RTE. Previous research has predominantly used the P1 approximation [30] and six-flux method [31] for solving radiation transfer in photocatalytic reactors. However, the P1 approximation method is known for its lower accuracy, and determining the phase function required in the Six Flux Model can be challenging, particularly for particle suspension systems. When compared to the previous studies on simulating CPC photoreactors, the DOM emerges as a favorable choice for addressing problems related to the absorption of non-uniformly distributed photocatalytic particles. This is attributed to its accuracy in RTE calculations involving anisotropic scattering media and its relatively low computational resource requirements. Hence, the DOM is chosen to solve the RTE inside the CPC photocatalytic reactor in this study. Additionally, the DOM is categorized based on the number of discrete

directions, such as S2 (8 directions), S4 (24 directions), S6 (48 directions), S8 (80 directions), S12 (168 directions), etc. In this study, the computation utilizes S4, representing 24 discrete spatial directions. The scattering coefficient σ and the absorption coefficient κ of the TiO$_2$ photocatalyst particles are listed in Table 2, which include two wavelength bands, namely UVA (320–400 nm) and UVB (275–320 nm), according to the literature [32]. It should be noted that these coefficients are all dimensionless coefficients, meaning they are multiplied by the concentration of particles to obtain the optical property parameters required in Equation (18). Since the absorption of TiO$_2$ photocatalyst primarily occurs in the ultraviolet region, these two wavelength bands cover all portions of solar radiation that TiO$_2$ can absorb.

**Table 2.** The absorption coefficient and scattering coefficient of titanium dioxide [32].

| Coefficient | Numerical Value/m$^2\cdot$kg$^{-1}$ |
|:---:|:---:|
| $\kappa^*_{UVA}$ | 189.9 |
| $\sigma^*_{UVA}$ | 1175.1 |
| $\kappa^*_{UVB}$ | 508.5 |
| $\sigma^*_{UVB}$ | 1016.1 |

### 3.1.4. Simulation Method and Process

In this study, COMSOL Multiphysics 5.4 was used to construct and simulate the complex process in the CPC photocatalytic reactor. Moreover, the software employs the finite element method (FEM) as its primary numerical technique, enabling accurate solutions to partial differential equations shown in this study. The flow chart of the simulation is shown in Figure 9.

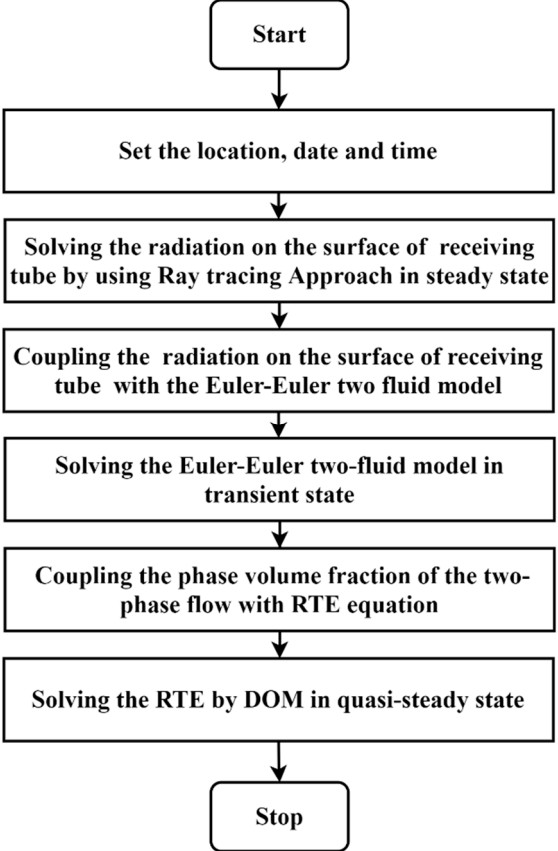

**Figure 9.** Flow chart of the simulation of CPC photocatalytic reactor in this study.

## 4. Conclusions

This study developed a comprehensive simulation model of a CPC tubular photocatalytic reactor under real outdoor operating conditions in Xi'an, Shaanxi Province, China, at 12:00 p.m. on 1 July (Beijing time). Initially, the concentrating performance of the CPC was analyzed using a ray tracing method. Results showed a non-uniform distribution of radiation intensity on the receiving tube surface, with the highest intensity reaching 7.6 times the incident solar radiation. Subsequently, a Euler–Euler two-fluid flow model was established, coupling the solar radiation received on the tube surface into the energy equation as a boundary heat source. Finally, the RTE in the receiving tube was solved by conducting the DOM. This computational model enabled the study of natural convection particle flow inside the receiving tube through transient calculations over a 120 s duration. The results also showed that natural circulation flow could not prevent the segregation of photocatalyst particles during the study period. Specifically, at the L = 1.8 m cross-section, the particle phase volume fraction decreased to 0 over time, while at other cross-sections (like L = 0.2 m, 0.6 m, 1.0 m, and 1.4 m), a stratified distribution of particle phase volume fraction was observed. Furthermore, the TRPA inside the tube revealed that at t = 120 s, the absorption of photocatalysts peaked, indicating that some level of segregation could enhance solar radiation absorption via the photocatalysts. Therefore, it was concluded that a mode dominated by natural circulation with intermittent disturbances is a rational and efficient approach for operating such reactors.

**Supplementary Materials:** The following supporting information can be downloaded at: https://www.mdpi.com/article/10.3390/catal14040237/s1. Including the details about CPC, the details of ray tracing method, analysis of the optic efficiency of CPC and introduction of the corresponding experimental platform in Word format, and some animations of the distributions of the temperature and phase volume fraction inside the receiving tube in GIF format.

**Author Contributions:** Conceptualization, B.L.; methodology, Q.W.; software, J.G.; validation, Q.W., S.Z. and B.L.; formal analysis, T.D.; investigation, S.Z. and T.D.; resources, L.M.; data curation, J.G.; writing—original draft preparation, J.G.; writing—review and editing, Y.L. and C.Z.; visualization, S.Z. and B.L.; supervision, C.Z.; project administration, Y.L.; funding acquisition, J.G., S.Z. and Y.L. All authors have read and agreed to the published version of the manuscript.

**Funding:** This research was funded by Young Scientists Fund of the National Natural Science Foundation of China, grant number: No. 22202016, Natural Science Basic Research Program of Shaanxi, grant number: No. 2023-JC-QN-0618, Fundamental Research Funds for the Central Universities, CHD: No.300102293104, 300102292107. The APC was funded by Young Scientists Fund of the National Natural Science Foundation of China, grant number: No. 22202016.

**Data Availability Statement:** The original contributions presented in the study are included in the article/supplementary material, further inquiries can be directed to the corresponding authors.

**Conflicts of Interest:** The authors declare no competing financial conflicts of interest.

## Nomenclature

*Symbols*

| | |
|---|---|
| $c$ | Speed of light |
| $c_{cat}$ | Concentration of the photocatalyst |
| $c_P$ | Specific heat capacity |
| $d_R$ | Diameter of the receiving tube |
| *Gap* | Gap between CPC and receiving tube |
| $k$ | Heat conductivity coefficient |
| $l_R$ | Length of the receiving tube |
| $n$ | Refractive index |
| $p$ | Pressure |
| $r_B$ | Radius of the base circle of the CPC |
| $C_D$ | Drag coefficient |
| $I$ | Radiation density |

| | |
|---|---|
| T | Temperature |
| Q | Quantity of heat |
| *Abbreviations* | |
| CPC | Compound parabolic concentrator |
| CFD | Computational fluid dynamics |
| DOM | Discrete ordinates method |
| LVRPA | Local volume radiative power absorption, $W \cdot m^{-3}$ |
| RTE | Radiative transfer equation |
| TRPA | Total radiative power absorption, W |
| UV | Ultraviolet |
| *Vectors and tensors* | |
| **F** | Volume forces, $N \cdot m^{-3}$ |
| **Fm** | Interphase interaction force, $N \cdot m^{-3}$ |
| **g** | Gravitational acceleration vector |
| **I** | Unit tensor |
| **k** | Wave vector |
| **q** | Position vector of the ray |
| **r** | Position vector of the space |
| **s** | Spatial angle vector |
| **u** | Velocity vector |
| **K** | Viscous stress tensor, Pa |
| **S** | Strain rate tensor |
| *Greek letters* | |
| $\alpha_{ht}$ | Efficiency of solar-to-thermal conversion |
| $\alpha_P$ | Thermal diffusion coefficient |
| $\varepsilon$ | Emission coefficient |
| $\theta_a$ | Maximum acceptance half angle, ° |
| $\theta_t$ | Truncation angle, ° |
| $\kappa$ | Absorption coefficient in aqueous medium, $m^{-1}$ |
| $\rho$ | Density, $kg \cdot m^{-3}$ |
| $\sigma$ | Scattering coefficient, $m^{-1}$ |
| $\tau$ | Viscous stress tensor |
| $\varphi$ | Azimuth angle in spherical coordinates |
| $\Psi$ | Polar angle in spherical coordinates |
| $\varphi$ | Phase volume fraction |
| $\varphi_S$ | Scattering phase function |
| $\Omega$ | Solid angle; its differential form is $d\Omega = \sin \psi d\psi d\varphi$, sr |
| *Superscripts and subscripts* | |
| c | Continuous phase |
| d | Discrete phase |
| m | Mixed phase |
| p | Particle |

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
