# Peer review of "A Numerical Case Study of Particle Flow and Solar Radiation Transfer in a Compound Parabolic Concentrator (CPC) Photocatalytic Reactor for Hydrogen Production"

_catalysts, doi:10.3390/catal14040237_

Round 1

Reviewer 1 Report

Comments and Suggestions for Authors

Compound parabolic concentrator has been gaining ever-increasing attention from academic researchers and industrial developers owing to its stationary feature for solar energy collection with a higher efficiency. The manuscript can be published in the journal. But at the same time, вased on the date given in the manuscript I have a few questions and several recommendations which can be considered by authors and it may be improved their work.

1. It is necessary to take into account the geometry of the compound parabolic concentrator, in which light losses and edge effects, which are determined by the geometry of the CPC, may occur. It is also necessary to take into account whether this leads to a decrease in optical efficiency?

Is it possible to reuse escaped light?

2. Does rotation occur? If not, can this be taken into account when constructing the model?

3. Will temperature (ts thermal performance) and electrolyte flow regime affect the efficiency of the hydrogen production process?

4. How will different weather conditions influence, can this be taken into account when building the model?

5. It is necessary to provide a more detailed description of the platform that was used in the design (possible in supplimentary materials), what characteristics it has.

Author Response

Reviewer #1:

Comment 1: It is necessary to take into account the geometry of the compound parabolic concentrator, in which light losses and edge effects, which are determined by the geometry of the CPC, may occur. It is also necessary to take into account whether this leads to a decrease in optical efficiency? Is it possible to reuse escaped light?

Response: Thank you very much for your kindness comments. We have added a supporting information (part S1) to discuss the issues regarding the geometry of CPC and its concentrating efficiency. Additionally, suggestions have been provided for further utilizing the escaped light.

Comment 2: Does rotation occur? If not, can this be taken into account when constructing the model?

Response: As we demonstrated in the supporting information, part S1, the model in our case study was established according to a real experimental platform, and the strategy of the tracking mode of the experimental setup is intermittent operation mode, which adjust 5°every hour. However, we only discussed the situation at a fixed time namely noon at 12 p.m., due to the high cost of commutating resource. But we will carefully consider the reviewer's comments and address them in our subsequent work.

Comment 3: Will temperature (ts thermal performance) and electrolyte flow regime affect the efficiency of the hydrogen production process?

Response: Temperature and flow pattern have a significant impact on the hydrogen production process, as discussed in our previous work (Ren Yuxun et al. Int. J. of Hydrog. Energy, 2016,41(36): 16019-16031. & Ma Lijing et al, Int. J. Hydrog. Energy, 2021, 46(3), 2871-2877.). Therefore, this study considered the temperature variations caused by solar radiation and the changes in particle distribution due to flow regime. In our case study, the temperature increases by up to 5°C, which was described in line 212. Furthermore, Furthermore, due to the non-uniform temperature distribution, natural convection occurred, leading to the formation of particle distribution within the tube, which further influenced the distribution of radiation energy and the rate of hydrogen production. Our case study effectively demonstrated the variations in temperature field, flow field, and particle concentration field induced by solar radiation, as well as their impact on the process of radiant energy transfer. However, a limitation of our study is the lack of correlation with the kinetic model of hydrogen production reaction. Further quantitative analysis will involve the kinetic model for hydrogen production, which will be very helpful for developing our future research.

Comment 4: How will different weather conditions influence, can this be taken into account when building the model?

Response: Weather conditions play a crucial role in hydrogen production under direct solar radiation. The impact is twofold: the difference in solar radiation intensity between sunny and cloudy days is significant, and environmental temperature affects the temperature distribution in the receiving tube, thus influencing hydrogen production efficiency. While this study primarily focuses on the Multiphysics modeling and multi-field coupling mechanisms of hydrogen production reactors under direct solar radiation, we briefly touch upon weather conditions. Additional explanations can be found in the manuscript, particularly in lines 211, 406-409, and 479-480. We are grateful for the valuable suggestions provided by the reviewers, which have steered our future research endeavors.

Comment 5: It is necessary to provide a more detailed description of the platform that was used in the design (possible in supplementary materials), what characteristics it has.

Response: We have provided details of the experimental platform in Supplementary Information Part S2.

Reviewer 2 Report

Comments and Suggestions for Authors

This work is the simulation of a compound parabolic concentrator containing solid photocatalyst particles inside the receiving pipe. The work mainly focuses on determining the thermal distribution inside the receiving pipe and catalyst particle distribution due to the convection flow inside the pipe. This work offers a better understanding of the physical processes in CPC photocatalytic reactors and provides theoretical support for their design, optimization, and operation. Some suggestions are given below:

(1)  The author should explain how the radiation was measured in the research.

(2) The paper does not compare or reference previous work related to the simulation of CPCs. The authors should include a comparison with earlier work on this topic.

(3)  The conclusion section needs to be shorter. Please shorten it.

(4)  There are some instances of errors in the English language. Please revise the paper carefully.

Comments on the Quality of English Language

The paper needs careful editing to improve the quality of English.

Author Response

Reviewer #2:

Comment 1: The author should explain how the radiation was measured in the research.

Response: Thanks for reviewer’s kindness comments. We have added a supporting information (part S1.2) to discuss the measurement of the radiation.

Comment 2: The paper does not compare, or reference previous work related to the simulation of CPCs. The authors should include a comparison with earlier work on this topic.

Response: The comparison with previous studies, as demonstrated in lines 489-492 and 501-509, has been included in our research. Our model focuses on natural convection driven by solar radiation and utilizes the Discrete Ordinates Method (DOM) to analyze the radiation distribution within the reactor. This helps to clarify the impact of radiation and particles in the reactor, along with relevant influencing factors. We value the reviewer's comment in improving the article's key points to enhance its readability.

Comment 3: The conclusion section needs to be shorter. Please shorten it.

Response: We have shortened the conclusion section from 491 words to 238 words, which has been marked in line 531-588.

Comment 4: There are some instances of errors in the English language. Please revise the paper carefully.

Response: We have carefully revised the spelling, grammar, and other aspects of the entire manuscript.

Round 2

Reviewer 1 Report

Comments and Suggestions for Authors

As I noted in a previous review, the manuscript can be accepted for publication. But in the future I would like to see experimental confirmation of the proposed model of a photoctalytic reactor and what photocatalyst will be used in this case. It should be noted that the model does not take into account the efficiency of the photocatalyst and how such a design affects the quantum yield. The authors provided answers to all my comments and they are well integrated into the text of the manuscript.